# *Alasemenia*, the earliest ovule with three wings and without cupule

Deming Wang[1]*[†], Jiangnan Yang[1][†], Le Liu[2][†], Yi Zhou[3], Peng Xu[1], Min Qin[4]*, Pu Huang[5]

[1]Key Laboratory of Orogenic Belts and Crustal Evolution, Department of Geology, Peking University, Beijing, China; [2]School of Geoscience and Surveying Engineering, China University of Mining and Technology, Beijing, China; [3]School of Life Sciences, Sun Yat-Sen University, Guangzhou, China; [4]College of Life Sciences, Linyi University, Linyi, China; [5]Nanjing Institute of Geology and Palaeontology, Chinese Academy of Sciences, Nanjing, China

**\*For correspondence:**
dmwang@pku.edu.cn (DW);
qinmin1990@yeah.net (MQ)

[†]These authors contributed equally to this work

**Competing interest:** The authors declare that no competing interests exist.

**Abstract** The ovules or seeds (fertilized ovules) with wings are widespread and especially important for wind dispersal. However, the earliest ovules in the Famennian of the Late Devonian are rarely known about the dispersal syndrome and usually surrounded by a cupule. From Xinhang, Anhui, China, we now report a new taxon of Famennian ovules, *Alasemenia tria* gen. et sp. nov. Each ovule of this taxon possesses three integumentary wings evidently extending outwards, folding inwards along abaxial side and enclosing most part of nucellus. The ovule is borne terminally on smooth dichotomous branches and lacks a cupule. *Alasemenia* suggests that the integuments of the earliest ovules without a cupule evolved functions in probable photosynthetic nutrition and wind dispersal. It indicates that the seed wing originated earlier than other wind dispersal mechanisms such as seed plume and pappus, and that three- or four-winged seeds were followed by seeds with less wings. Mathematical analysis shows that three-winged seeds are more adapted to wind dispersal than seeds with one, two or four wings under the same condition.

## eLife assessment

This **useful** study describes the second earliest known winged ovule without a capule in the Famennian of Late Devonian. Using **solid** mathematical analysis, the authors demonstrate that three-winged seeds are more adapted to wind dispersal than one-, two- and four-winged seeds. The manuscript will help the scientific community to understand the origin and early evolutionary history of wind dispersal strategy of early land plants.

## Introduction

Since plants colonized the land, wind dispersal (anemochory) became common with the seed wing representing a key dispersal strategy through geological history (*Taylor et al., 2009*; *Ma, 2009*; *McLoughlin and Pott, 2019*). Winged seeds evolved numerous times in many lineages of extinct and extant seed plants (spermatophytes) (*Schenk, 2013*; *Stevenson et al., 2015*). Lacking wings as integumentary outgrowths, the earliest ovules in the Famennian (372–359 million years ago [Ma], Late Devonian) rarely played a role in wind dispersal (*Rowe, 1997*). Furthermore, nearly all Famennian ovules are cupulate, i.e., borne in a protecting and pollinating cupule (*Prestianni et al., 2013*; *Meyer-Berthaud et al., 2018*).

**eLife digest** Many plants need seeds to reproduce. Seeds come in all shapes and sizes and often have extra features that help them disperse in the environment. For example, some seeds develop wings from seed coat as an outer layer, similar to fruits of sycamore trees that have two wings to help them glide in the wind.

The first seeds are thought to have evolved around 372-359 million years ago in a period known as the Famennian (belonging to the Late Devonian). Fossil records indicate that almost all these seeds were surrounded by an additional protective structure known as the cupule and did not have wings. To date, only two groups of Famennian seeds have been reported to bear wings or wing-like structures, and one of these groups did not have cupules. These Famennian seeds all had four wings.

Wang et al. examined fossils of seed plants collected in Anhui province, China, which date to the Famennian period. The team identified a new group of seed plants named the *Alasemenia* genus. The seeds of these plants each had three wings but no cupules. The seeds formed on branches that did not have any leaves, which indicates the seeds may have performed photosynthesis (the process by which plants generate energy from sunlight). Mathematical modelling suggested that these three-winged seeds were better adapted to being dispersed by the wind than other seeds with one, two or four wings.

These findings suggest that during the Famennian the outer layer of some seeds that lacked cupules evolved wings to help the seeds disperse in the wind. It also indicates that seeds with four or three wings evolved first, followed by other groups of seed plants with fewer seed wings. Future studies may find more winged seeds and further our understanding of their evolutionary roles in the early history of seed plants.

*Warsteinia* was a Famennian ovule with four integumentary wings, but its attachment and cupule remain unknown (*Rowe, 1997*). *Guazia* was a Famennian ovule with four wings and it is terminally borne and acupulate (devoid of cupule) (*Wang et al., 2022*). This paper documents a new Famennian seed plant with ovule, *Alasemenia tria* gen. et sp. nov. It occurs in Jianchuan mine of China, where Xinhang fossil forest was discovered to comprise *in situ* lycopsid trees of *Guangdedendron* (*Wang et al., 2019*). The terminally borne ovules are three-winged and clearly acupulate, thus implying additional or novel functions of integument. Based on current fossil evidence and mathematical analysis, we discuss the evolution of winged seeds and compare the wind dispersal of seeds with different number of wings.

## Results

### Locality, stratigraphy, and material

All fossils came from Upper Devonian Wutong Formation at Jianchuan mine in Xinhang town, Guangde City, Anhui Province, China. Details on locality are available in previous works (*Wang et al., 2019*; *Xu et al., 2022*). At Jianchuan mine, Wutong Formation consists of Guanshan Member with quartzose sandstone and a little mudstone, and the overlying Leigutai Member with inter-beds of quartzose sandstone, siltstone and mudstone. Spore analysis indicates that the Leigutai Member here is late Famennian in age (*Gao et al., 2023*). Progymnosperm *Archaeopteris* and lycopsid *Leptophloeum* occur in Leigutai and/or Guanshan members, and they were distributed worldwide in the Late Devonian (*Taylor et al., 2009*). Fernlike plant *Xinhangia* (*Yang and Wang, 2022*) and lycopsid *Sublepidodendron* (*Xu et al., 2022*) were found at the basal part of Leigutai Member. *In situ* lycopsid trees of *Guangdedendron* with stigmarian rooting system appear in multiple horizons of Leigutai Member and they formed the Xinhang forest (*Wang et al., 2019*; *Gao et al., 2022*). From many horizons of siltstone and mudstone of Wutong Formation (Leigutai Member) at Jianchuan mine, numerous ovules of *Alasemenia* were collected (*Figure 1*, *Figure 2*, *Figure 3*) and some were transversely sectioned to show the ovular structure (*Figure 4*, *Figure 4—figure supplement 1*, *Figure 4—figure supplement*

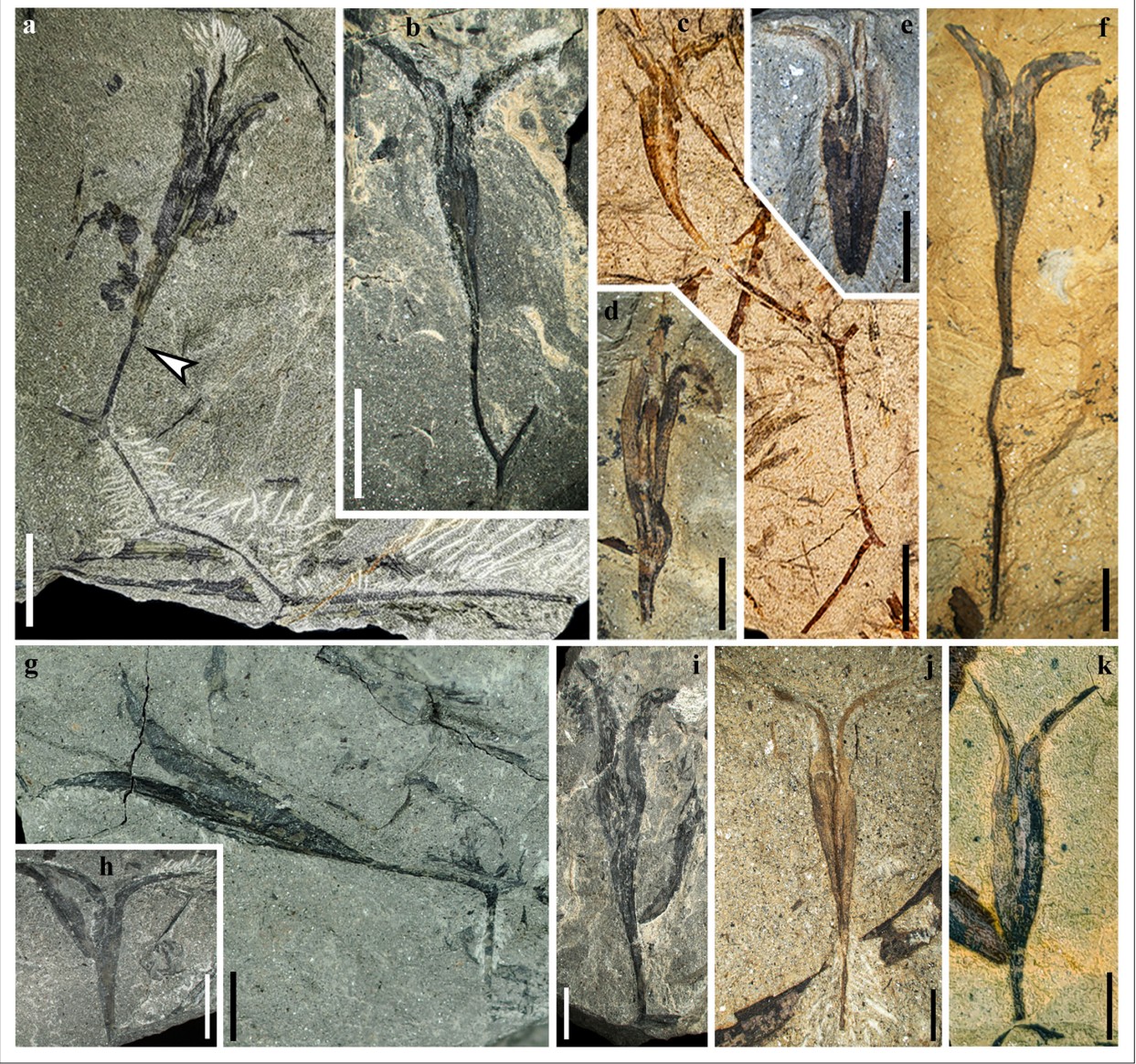

**Figure 1.** Fertile branches and seeds of *Alasemenia tria* gen. et sp. nov. (**a**) Thrice dichotomous branch with a terminal ovule. Arrow indicating boundary between ovule and ultimate axis (PKUB21721a). (**b, f, g, i**) Once dichotomous branch with a terminal ovule (PKUB21781, PKUB23132, PKUB19338a, PKUB17899). (**c**) Twice dichotomous branch with a terminal ovule (PKUB19713a). (**d, e**) Ovule with three integumentary wings (PKUB19321, PKUB19316). (**h**) Ovule showing two integumentary wings (PKUB19282). (**j, k**) Ovule terminating short ultimate axis (PKUB23114, PKUB23129). Scale bars, 1 cm (**a–c, h**), 5 mm (**d–g, i–k**).

*2*, *Figure 4—figure supplement 3*, *Figure 4—figure supplement 4*, *Figure 4—figure supplement 5*).

## Systematic palaeontology

Division Spermatophyta
Order and family incertae sedis
*Alasemenia tria* gen. et sp. nov.

## Etymology

The generic name from the Latin 'ala' and 'semen', meaning wing and seed, respectively; the specific epithet from the Latin 'tri' (three), referring to wing number of a seed.

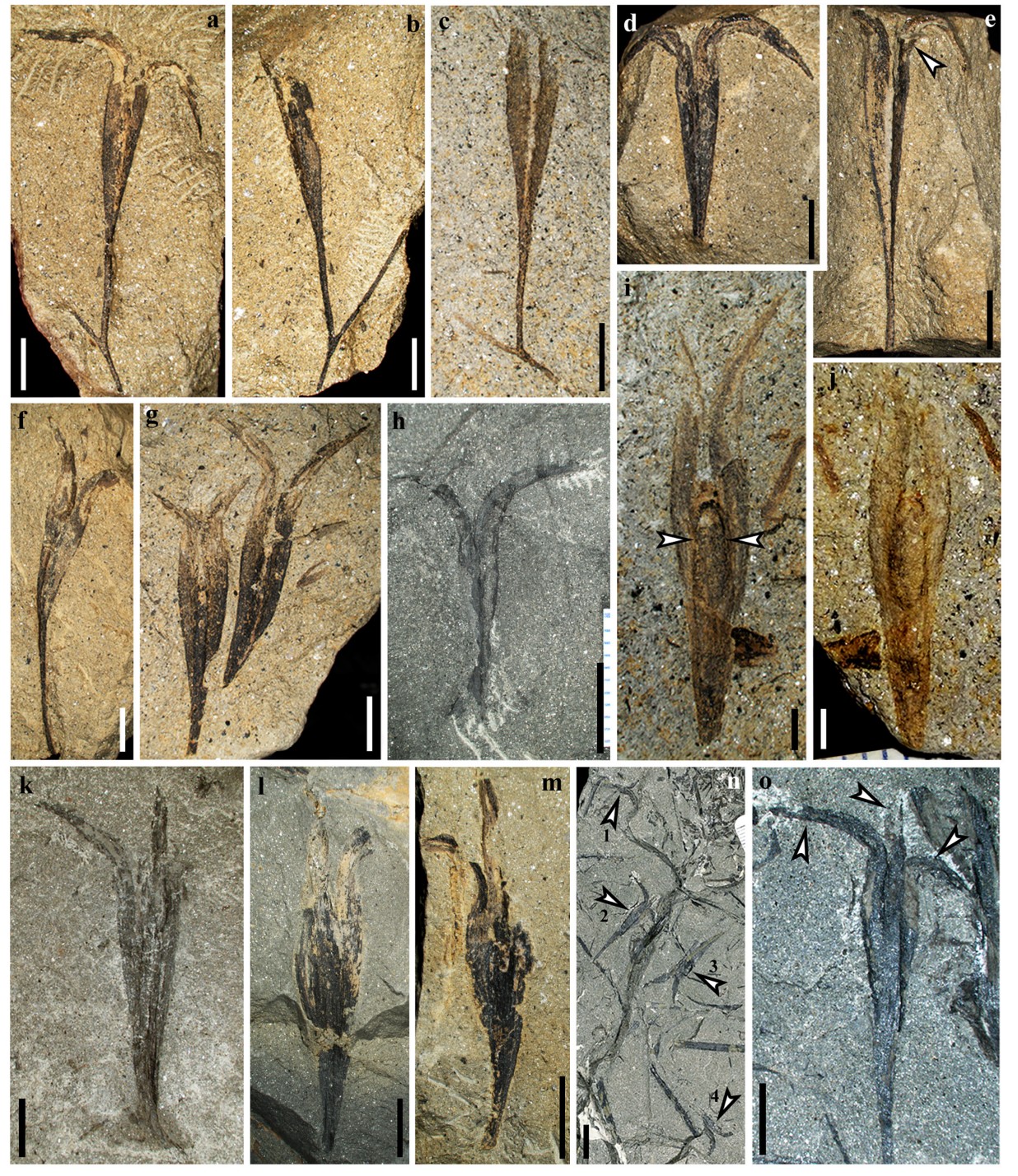

**Figure 2.** Fertile branches and seeds of *Alasemenia tria* gen. et sp. nov.    (**a–c**) Once dichotomous branch with a terminal ovule (PKUB16876a, b, PKUB17767). a, b, Part and counterpart. (**d, e**) Part and counterpart, arrow showing the third integumentary wing (PKUB19322a, b). (**f**) Ovule on ultimate axis (PKUB21752). (**g, h, k–m**) Ovules lacking ultimate axis (PKUB16788, PKUB21631, PKUB16522, PKUB21647, PKUB21656). (**i, j**) Part and counterpart, showing limit (arrows) between nucellus and integument (PKUB19339a, b). (**n**) Four detached ovules (arrows 1–4) (PKUB19331). (**o**) Enlarged ovule in n (arrow 2), showing three integumentary wings (arrows). Scale bars, 1 cm (**n**), 5 mm (**a–h, k–m, o**), 2 mm (**i, j**).

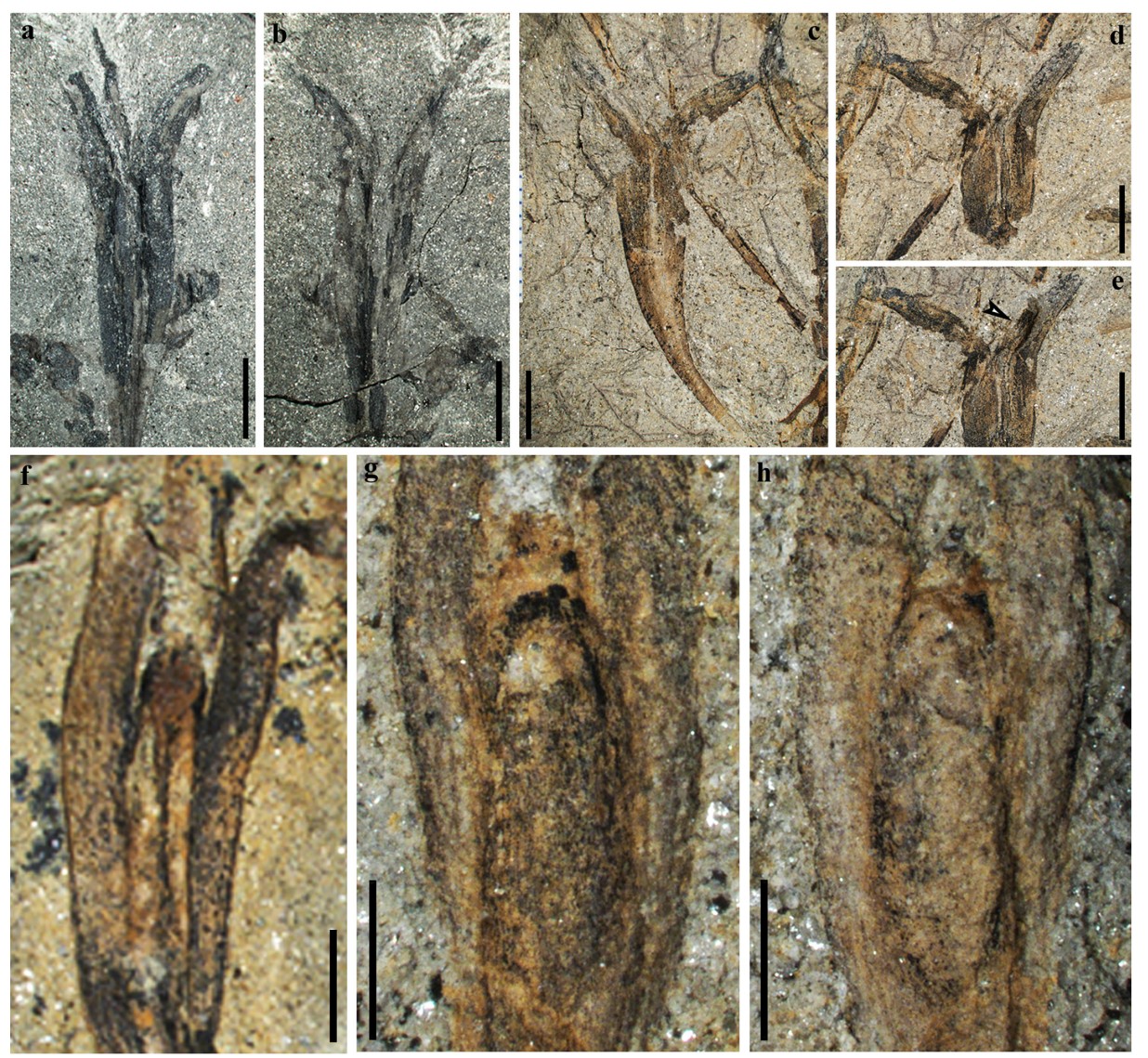

**Figure 3.** Seeds of *Alasemenia tria* gen. et sp. nov. (**a, b**) Part and counterpart, enlarged ovule in *Figure 1a* (PKUB21721a, b). (**c**) Enlarged ovule in *Figure 1c*. (**d**) Counterpart of ovule in c (PKUB19713b). (**e**) Dégagement of ovule in d, exposing the base of the third integumentary wing (arrow). (**f**) Enlarged ovule in *Figure 1d*. (**g, h**) Enlarged ovule in *Figure 2i and j*, respectively. Scale bars, 5 mm (**a–e**), 2 mm (**f–h**).

## Holotype designated here

PKUB21721a, b (part and counterpart housed in Department of Geology, Peking University, Beijing) (*Figure 1a, m and n*).

## Locality and horizon

Xinhang, Guangde, Anhui, China; Leigutai Member of Wutong Formation, Upper Devonian.

## Diagnosis

Dichotomous branches bearing terminal and acupulate ovules. Three broad wing-like integumentary lobes radially and symmetrically attached to each nucellus, distally tapered and proximally reduced. Integumentary lobes evidently extending outwards, with their free parts ca. 40% of ovule length. Individual integumentary lobes folding inwards along abaxial side. Nucellus largely adnate to integument.

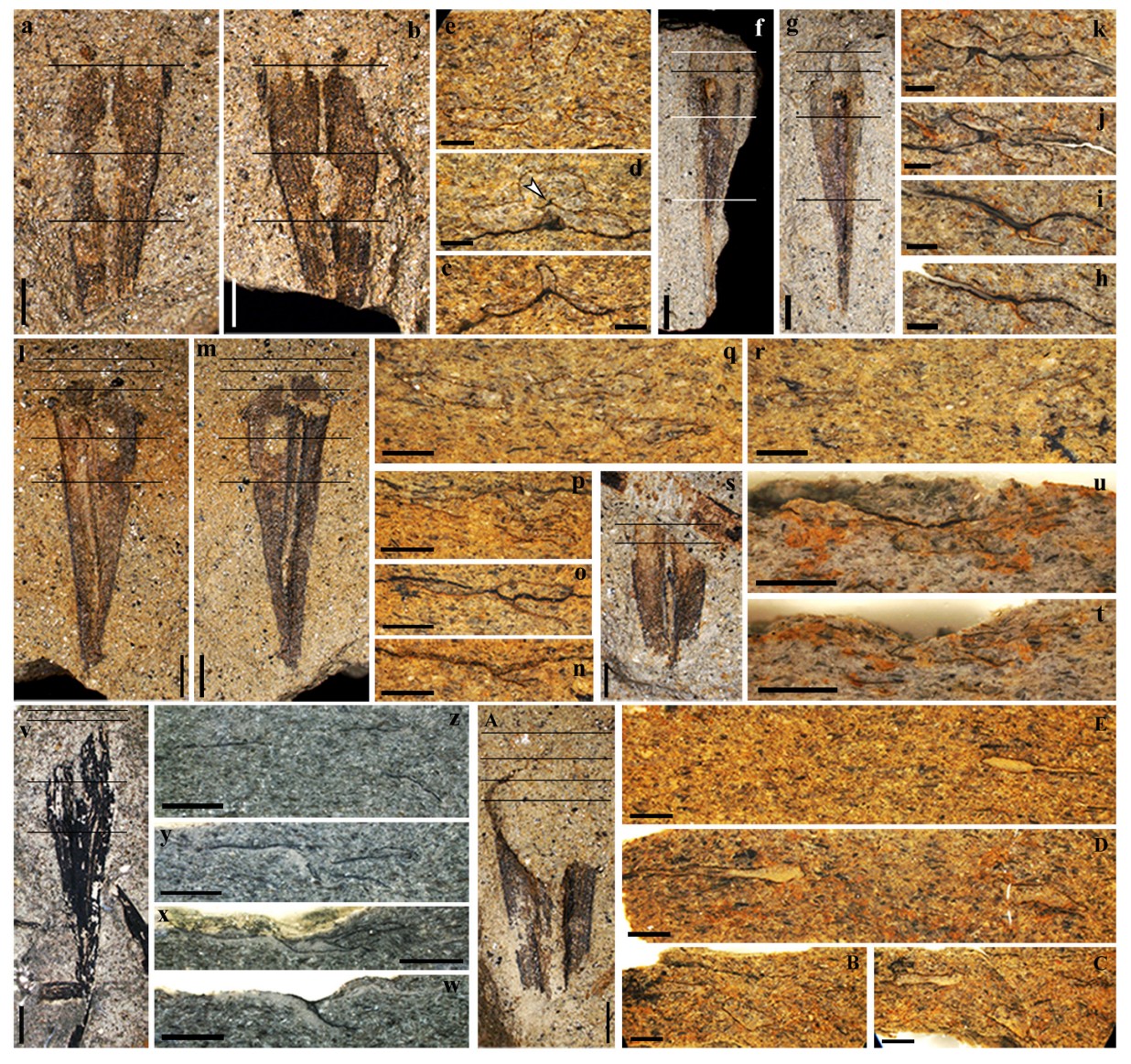

**Figure 4.** Transverse sections of seeds of *Alasemenia tria* gen. et sp. nov. (**a, b**) Part and counterpart. (**c–e**) Sections of seed in a and b (at three lines, in ascending orders). Arrow in d indicating probable nucellar tip (Slide PKUBC17913-12b, 10a, 9b). (**f, g**) Part and counterpart. (**h–k**) Sections of seed in f and g (at four lines, in ascending orders) (Slide PKUBC19798-8b, 6b, 4a, 4b). (**l, m**) Part and counterpart. (**n–r**) Sections of seed in l and m (at five lines, in ascending orders), showing three wings departing centrifugally (Slide PKUBC17835-5a, 7b, 8b, 9a, 10a). (**s, v, A**), One seed sectioned. (**t, u**) Sections of seed in s (at two lines, in ascending orders) (Slide PKUBC18716-8b, 7a). (**w–z**) Sections of seed in v (at four lines, in ascending orders) (Slide PKUBC20774-7a, 6b, 3a, 3b). (**B–E**) Sections of seed in A (at four lines, in ascending orders), showing three wings departing centrifugally (Slide PKUB17904-5b, 4a, 4b, 3b). Scale bars, 2 mm (**a, b, f, g, l, m, s, v, A**), 1 mm (**c–e, h–k, n–r, t, u, w–z, B–E**).

The online version of this article includes the following figure supplement(s) for figure 4:

**Figure supplement 1.** Transverse sections of two seeds of *Alasemenia tria* gen. et sp. nov.

**Figure supplement 2.** Transverse sections of one seed of *Alasemenia tria* gen. et sp. nov.

**Figure supplement 3.** Transverse sections of two seeds of *Alasemenia tria* gen. et sp. nov.

**Figure supplement 4.** Transverse sections of one seed of *Alasemenia tria* gen. et sp. nov.

**Figure supplement 5.** Transverse sections of one seed of *Alasemenia tria* gen. et sp. nov.

## Description

Some ovules are borne terminally on smooth branches that are thrice (*Figure 1a*), twice (*Figure 1c*) or once (*Figure 1b, f, g and i*, *Figure 2a–c*) dichotomous at 40–135°. The boundary between ovule and ultimate axis below refers to position where the ovule width just begins to increase (e.g. *Figure 1a*, arrow). The branches excluding ovules are up to 76 mm long and 0.4–0.9 mm wide. Most ovules terminate ultimate axis (*Figure 1j*, *Figure 2d–f*) or are detached (*Figure 1d, e, h and k*, *Figure 2g–o*). The ovules are 25.0–33.0 mm long and 3.5–5.6 mm at the maximum width (excluding the width of outward extension of integumentary wings). Compressions of ovules (*Figure 1*, *Figure 2*, *Figure 3*) and their serial transverse sections (*Figure 4*, *Figure 4—figure supplement 1*, *Figure 4—figure supplement 2*, *Figure 4—figure supplement 3*, *Figure 4—figure supplement 4*, *Figure 4—figure supplement 5*) do not show any cupules.

Each ovule possesses a layer of integument with three radially arranged and wing-like integumentary lobes (*Figure 1a, d and e*, *Figure 2d, e and o*, arrows, *Figure 3a–f*, *Figure 4*, *Figure 4—figure supplement 1*, *Figure 4—figure supplement 2*, *Figure 4—figure supplement 3*, *Figure 4—figure supplement 4*, *Figure 4—figure supplement 5*). They are broad, acropetally tapered and proximally reduced to merge with the ultimate axis (*Figure 1a–c and f*, *Figure 2a, d–i and n*). The integumentary lobes are 1.2–2.3 mm at the maximum width and free for 8.3–14.8 mm distance (32–45% of the ovule length), and the free lobe parts extend well above the nucellar tip and greatly curve outward. Usually, two lobes of a single ovule are evident and the third one is sometimes exposed through dégagement (*Figure 1a*, *Figure 2n*, arrow 2, o, middle arrow, *Figure 3a and e*, arrow). Such situation indicates that the lobes of an ovule are present on different bedding planes.

In transverse sections of an ovule, two integumentary lobes extend along the bedding plane, and the third lobe is either originally perpendicular to or compressed to lie somewhat along the bedding plane (*Figure 4c–e, h–k, n–r, t and w–y*, *Figure 4—figure supplement 1*, *Figure 4—figure supplement 2*, *Figure 4—figure supplement 3*, *Figure 4—figure supplement 4*, *Figure 4—figure supplement 5*). The integumentary lobes are narrow, flattened and fused in the lower part of an ovule, and acropetally become wide, thick, separated and far away (*Figure 4c–e*, *Figure 4—figure supplement 1a–k*, *Figure 4—figure supplement 2*). Because of great outward curving of lobes, it is difficult to observe their distal parts in the sections. When thick, the lobes present a V or U shape. Therefore, they are symmetrically folded along the abaxial side and toward the ovule center.

A few ovules show the outline of a nucellus, which is ca. 10–11.7 mm long and 1.2–1.7 mm at the maximum width (*Figures 1d and 2i*, arrows, j, *Figure 3f–h*). Transverse sections occasionally meet the

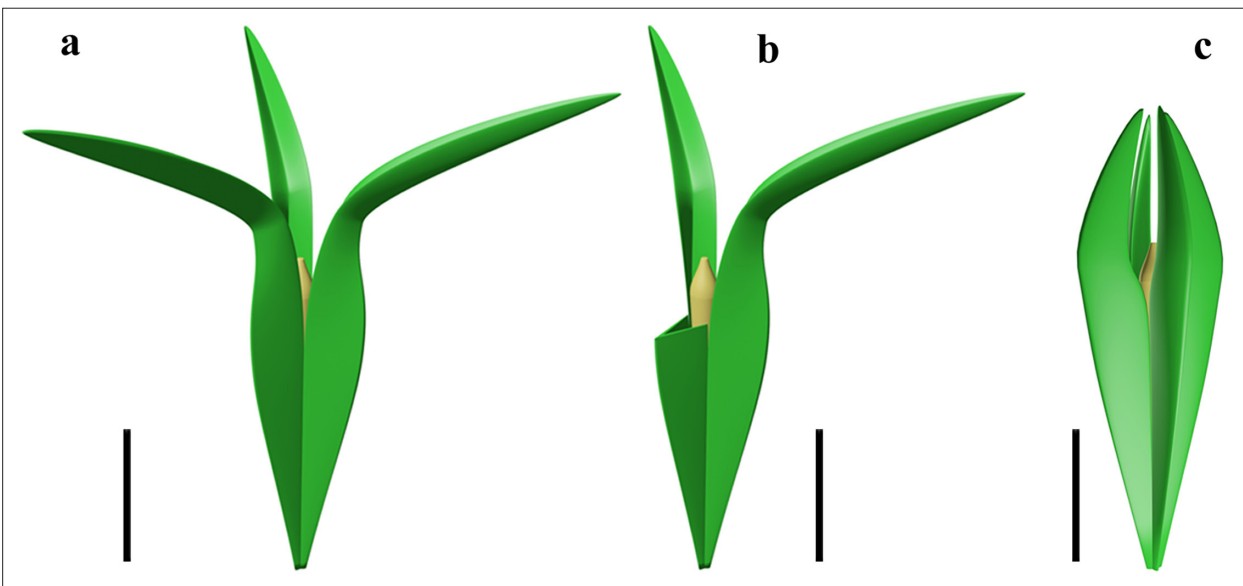

**Figure 5.** Reconstruction of two acupulate ovules with integumentary wings. (**a**) *Alasemenia tria* with three wings distally extending outwards. (**b**), *A. tria* with one of three wings partly removed to show nucellar tip. (**c**) *Guazia dongzhiensis* with four wings distally extending inwards (*Wang et al., 2022*). Scale bars, 5 mm.

nucellar tip (*Figure 4d*, arrow). With the exception of tip, the nucellus is adnate to the integument and is distally surrounded by the free parts of integumentary lobes. The ovule is reconstructed to show three integumentary wings (*Figure 5a*) and nucellar tip (*Figure 5b*).

## Discussion

### Late Devonian acupulate ovules and their functions

Except few taxa including *Guazia* (*Wang et al., 2022*), *Dorinnotheca* (*Fairon-Demaret, 1996*), *Cosmosperma* (*Liu et al., 2017*), *Elkinsia* (*Serbet and Rothwell, 1992*) and *Moresnetia* (*Fairon-Demaret and Scheckler, 1987*), the earliest ovules in the Late Devonian (Famennian) have not been preserved to be connected to the branches. These ovules are usually surrounded by cupules. Now, as in *Guazia*, the ovules of *Alasemenia* terminate the dichotomous branches and are transversely sectioned to show the ovular structures. The ovules of these two genera provide rare evidence for acupulate ovules in the earliest seed plants.

Both cupule and integument of an early ovule perform the protective and pollinating functions (Meyer-Berthaud, 2022). The integument of *Alasemenia* is adnate to most part of the nucellus and its three lobes extend long distance above the nucellar tip and then evidently outwards. Such structure leads to efficient protection of nucellus and adaptation for pollination. In early seed plants, the fertile branches terminated by cupulate ovules consistently lack leaves and thus the cupules probably serve a nutritive function as in photosynthetic organs; this function may be transferred to the integuments of acupulate ovules such as *Guazia* (Meyer-Berthaud, 2022). Regarding *Alasemenia*, the nutritive function would also apply to the integuments since the acupulate ovules are terminal on various orders of naked branches and ultimate axes. The surface of integuments is enlarged through the outgrowths of wings and thus promotes the photosynthesis. Little is known about the dispersal function of Famennian ovules, because the integumentary wings as a derived character have been rarely documented. *Warsteinia* (*Rowe, 1997*), *Guazia* (*Wang et al., 2022*) and now *Alasemenia* indicate that the anemochory originated in the Famennian. However, the ovule of *Warsteinia* remains unclear in attachment to branches and possession of a cupule. *Alasemenia* confirms that, like *Guazia*, the integuments of acupulate ovules developed a new function in wind dispersal.

### Late Devonian winged ovules and evolution of ovular wings

In the Late Devonian ovules, besides *Alasemenia*, both *Warsteinia* (*Rowe, 1997*) and *Guazia* (*Wang et al., 2022*) possess integumentary lobes forming radially arranged wings. Their integumentary wings illustrate diversity in number (three or four per ovule), length, folding or flattening, and being straight or curving distally. As in *Alasemenia* (*Figure 5a*), the integumentary wings of acupulate ovule of *Guazia* are broad, thin and fold inwards along the abaxial side, but their numbers are four in each ovule and their free portions usually arch centripetally (*Figure 5c*; *Wang et al., 2022*, Figure 5). In contrast to *Alasemenia*, *Warsteinia* has four integumentary wings without folding and their free portions are short and straight (*Rowe, 1997*, TEXT-FIGURE 4). Ovules of *Alasemenia* and *Guazia* terminating long and narrow branches suggest easy abscission of diaspores (ovules with or without an ultimate axis) and better preparation for dispersal. Compared to *Warsteinia* with short and straight wings and *Guazia* with long but distally inwards curving wings, *Alasemenia* with longer and outwards extending wings would efficiently reduce the rate of descent and be more capably moved by wind. Furthermore, the quantitative analysis in mathematics indicates that three-winged ovules such as *Alasemenia* are more adapted to wind dispersal than four-winged ovules including *Warsteinia* and *Guazia* (see following).

*Alasemenia*, *Guazia,* and *Warsteinia* suggest that the evolutionarily novel wings, as integument outgrowths and the most important mechanism for seed dispersal by wind, appeared early in the spermatophytes and had been manifested in younger lineages. Other wind dispersal mechanisms including plumes, pappi and parachutes of seeds appeared later in the Permo-Carboniferous and Mesozoic, respectively (*Axsmith et al., 2013*). Current evidence indicates that seeds with three or four wings occurred first in the Late Devonian. They were followed by two- or three-winged seeds in the Carboniferous (*Long, 1960*; *Long, 1969*), and then by single-winged seeds in the Permian (*Stevenson et al., 2015*; *Prevec et al., 2008*). Relating to wind dispersal, the diaspores (seeds/fruits) of living spermatophytes possess multiple mechanisms and variable number of wings (*Ma, 2009*).

## Mathematical analysis of wind dispersal of ovules with 1-4 wings

The rate of diaspore descent in still air is an important indicator of the potential ability of modern samaras dispersal (**Augspurger et al., 2016**; **Augspurger et al., 2017**). In examining samaras, it has been demonstrated that the value of angular velocity is smaller than that of terminal velocity (**Green, 1980**), and the relationship between the samaras' wing loading and terminal velocity $v_{ter}$ is:

$$v_{ter} \propto \sqrt{\frac{w}{A_w}}$$

where $w$ is the weight of samaras, $A_W$ is the surface area of the wing, $w/A_W$ is defined as the samaras' wing loading.

As for ovules, we also use terminal velocity as an indicator of dispersal ability. Since the broad integumentary wings well extend outwards, the wing loading of *Alasemenia* is obviously less than that of *Guazia*. When the winged seeds fall in the air, the predictable spinning can lead to the reduction of fall rate, and result in the increase of the horizontal dispersal distance. This has been observed in the field experiments or proved in the modelling reconstruction experiments (**Green, 1980**; **Habgood et al., 1998**).

However, the tiny asymmetry of ovules will be amplified in the running geometry by centrifugal and aerodynamic loads, result in vibrations and thus significantly reduce the efficiency (**Lu et al., 2019**). The transverse wave caused by vibrations will arrive the tip of wing and form stationary waves. In the ovules with even number of wings like *Guazia* (**Wang et al., 2022**) and *Warsteinia* (**Rowe, 1992**; **Rowe, 1997**), the center symmetry structure will lead to stronger resonance than the ovules with odd number of wings (like *Alasemenia*). It means the ovules with odd number of wings are more stable in high rate spinning and spend more falling time in the dispersal process.

Another consideration is the capacity of airflow in horizontal direction. The Reynolds number (Re) is the indicator of patterns in fluid flow situations, and for ovules, the Reynolds numbers are mainly falling in $10^3$–$10^4$, suggesting that the inertia forces are much stronger than viscous forces (**Burrows, 1975**; **Seter and Rosen, 1992**). In this situation, the thrust of airflow can be represented as:

$$F = c\rho v^2 S$$

| The number of wings (n) | Top view | Area of windward | | | $D(\theta)/S_{wing}$ | Relative efficiency $E_r(\%)$ |
| | | $S(\theta)$ | | | | |
| | | Function | Graph | | | |
| $n = 2$ <br> $n = 1$ <br> the wings keep facing the wind | airflow ↓↓↓ | $S(\theta) = 2S_{wing},$ <br> $S(\theta) = S_{wing}$ <br> $\theta \in [0, 2\pi].$ | | | $4\pi$ <br> $2\pi$ | 100.0 |
| $n = 2$ <br> $n = 1$ | airflow ↓↓↓ | $S(\theta) = 2\sin\theta\, S_{wing}$ <br> $S(\theta) = \sin\theta\, S_{wing}$ <br> $\theta \in [0, \pi]$ | | | $8$ <br> $4$ | 63.66 |
| $n = 3$ (*Alasemenia*) | airflow ↓↓↓ | $S(\theta) = \sqrt{3}S_{wing}\cos\left(\frac{\pi}{3} - \theta\right)$ <br> $\theta \in [\frac{\pi}{6}, \frac{\pi}{3}]$ | | | $6$ | 82.70 |
| $n = 4$ (*Guazia*) | airflow ↓↓↓ | $S(\theta) = 2S_{wing}\max(\sin\theta, \cos\theta)$ <br> $\theta \in [0, \pi/2]$ | | | $8\sqrt{2}$ | 90.03 |

**Figure 6.** The mathematical analysis of wind dispersal ability of ovules with 1–4 wings. The maximum windward area of each wing is $S_{wing}$ and $n$ represents the number of wings per ovule. $r$ is the distance from the tip of wing to the axis of ovules. $S(\theta)$ represents the area of windward when the angle between airflow and wings is $\theta$, and $D(\theta)$ represents the accumulated area of windward in a cycle. $E_r(\%)$ means relative efficiency for wind dispersal. Red lines and expressions show the situation of $n = 1$.

where c is the coefficient, $\rho$ is the density of air, $S$ is the windward face, $v$ is the velocity of relative movement.

It means that we can transform the comparison of capacity of airflow into the area of windward. Supposing the maximum windward area of each wing is $S_{wing}$ and $n$ represents the number of wings revolving on its own axis, we define a function $S(\theta)$ to represent the area of windward when the angle between airflow and wings is $\theta$. We introduce relative efficiency $E_r$ for comparison. Here, we list the following 5 ideal basic situations and the summary results are shown in **Figure 6**.

1. $n = 2$ and the wings keep facing the wind (wings without rotation, as a control group). In this situation, the area of windward is identical to $2S_{wing}$, which can be written as:

$$S(\theta) = 2S_{wing}, \theta \in [0, 2\pi],$$

$$(\text{if } n = 1,\ S(\theta) = S_{wing},\ \theta \in [0, 2\pi])$$

We define a function $D(\theta)$ to represent the accumulated area of windward in a cycle. By definition,

$$D(\theta) = 2\pi\, S(\theta) = 4\pi S_{wing},$$

$$E_r = 100\%$$

2. $n = 2$. In this situation, we discuss the condition when $\theta \in [0, \pi]$.

$$S(\theta) = 2\sin\theta\, S_{wing} \in [0, 2S_{wing}],$$

Based on symmetry,

$$D(\theta) = 2\int_0^\pi S(\theta)\, d\theta = 4S_{wing}\int_0^\pi \sin\theta d\theta = 4S_{wing}(\cos 0 - \cos\pi) = 8S_{wing},$$

$$E_r = 8S_{wing}/4\pi S_{wing} \approx 63.66\%$$

3. $n = 1$. In this situation,

$$S(\theta) = \sin\theta S_{wing} \in [0,\ S_{wing}],$$

$$D(\theta) = 4S_{wing},$$

As can be seen in **Figure 6**, the relative efficiency is equal to the situation of $n = 2$,

$$E_r \approx 63.66\%$$

4. $n = 3$ (*Alasemenia*).

$$S(\theta) = 2S_{wing}\left(\sin(\theta) + \sin(\frac{2\pi}{3} - \theta)\right) = 2S_{wing}\sin\left(\frac{\pi}{3}\right)\cos\left(\frac{\pi}{3} - \theta\right),$$

$$S(\theta) = \sqrt{3}S_{wing}\cos\left(\frac{\pi}{3} - \theta\right) \in \left[\frac{3S_{wing}}{2}, \sqrt{3}S_{wing}\right],\ \theta \in [\pi/6, \pi/3].$$

Based on symmetry,

$$D(\theta) = 12\int_{\frac{\pi}{6}}^{\frac{\pi}{3}} S(\theta)\, d\theta = 12\sqrt{3}S_{wing}\left(\sin\frac{\pi}{6} - \sin 0\right) = 6\sqrt{3}S_{wing},$$

$$E_r = 6\sqrt{3}S_{wing}/4\pi S_{wing} \approx 82.70\%$$

5. $n = 4$ (*Guazia, Warsteinia*).
When $\theta \in [\pi/4, \pi/2]$,

$$S(\theta) = 2S_{wing}\sin\theta \in \left[\sqrt{2}S_{wing}, 2S_{wing}\right]$$

Based on symmetry,

$$D\left(\theta\right) = 8 \times 2S_{wing} \int_{\frac{\pi}{4}}^{\frac{\pi}{2}} \sin\theta d\theta = 16S_{wing} \left(\cos\frac{\pi}{4} - \cos\frac{\pi}{2}\right) = 8\sqrt{2}S_{wing}$$

$$E_r = 8\sqrt{2}S_{wing}/4\pi S_{wing} \approx 90.03\%$$

Generally, the above one- to four-winged seeds are quantitatively analysed for their wind dispersal capability and the results are shown in *Figure 6*. The relative efficiency for wind dispersal ($E_r$) of these seeds in five ideal basic situations is calculated for comparison. The one- or two-winged seeds are treated as a control group when the wings keep facing the wind and do not rotate. In this case, the value of $E_r$ is 100%. When descending through autorotation, one- to four-winged seeds present different values of $E_r$. The relative wind dispersal efficiency of three-winged seeds is obviously better than that of single- and two-winged seeds, and is close to that of four-winged seeds (*Figure 6*). In addition, three-winged seeds have the most stable area of windward, which also ensures the motion stability in wind dispersal. Significantly, the maximum windward area of each wing of *Alasemenia* is greater than that of *Guazia* and *Warsteinia* with four wings. All these factors suggest that *Alasemenia* is well adapted for anemochory.

## Conclusion

The earliest ovules in the Famennian of Late Devonian, usually preserved to be unconnected with branches, are mostly devoid of integumentary wings and enclosed in cupules; they are thus insufficiently known for integument function and wind dispersal. New ovule *Alasemenia* in this paper terminates leafless branches, bears three wings and lacks cupules. It represents the second acupulate ovule in the Famennian. Besides protective and pollinating functions, *Alasemenia* suggests photosynthetically nutritive and anemochorous functions of early ovules. It indicates the diversity of Famennian winged ovules and the evolutionary sequence of ovule wings. Compared to Famennian four-winged ovules of *Warsteinia* and *Guazia*, *Alasemenia* with three distally outwards extending wings shows advantage in anemochory. Mathematical analysis implies that three-winged ovules could be efficiently dispersed by wind.

## Methods

All specimens are housed in Department of Geology, Peking University, Beijing, China. Steel needles were used to expose some seeds and fertile axes. Serial dégagement was employed to reveal the morphology and structure of seeds. Seeds were embedded in resin, sectioned and ground to show the integuments and nucelli. All photographs were made with a digital camera and microscope.

## Acknowledgements

We thank X Gao, Z Z Deng and L Liu for help in fieldwork. Q Y Jia and D B Ni provide assistance in the experiment for seed sections. This work was supported by the National Natural Science Foundation of China (grant no. 42130201).

## Additional information

### Funding

| Funder | Grant reference number | Author |
| --- | --- | --- |
| National Natural Science Foundation of China | 42130201 | Deming Wang |

The funders had no role in study design, data collection and interpretation, or the decision to submit the work for publication.

### Author contributions

Deming Wang, Conceptualization, Data curation, Formal analysis, Funding acquisition, Investigation, Writing – original draft, Writing – review and editing; Jiangnan Yang, Data curation,

Investigation; Le Liu, Conceptualization, Data curation, Investigation; Yi Zhou, Formal analysis, Writing – original draft; Peng Xu, Pu Huang, Investigation; Min Qin, Conceptualization, Project administration

### Author ORCIDs
Deming Wang ⓘ https://orcid.org/0000-0001-8334-2724
Le Liu ⓘ http://orcid.org/0000-0002-9099-6029
Yi Zhou ⓘ http://orcid.org/0000-0002-8910-0061
Min Qin ⓘ http://orcid.org/0000-0001-6088-8572

Reviewer #1 (Public Review): https://doi.org/10.7554/eLife.92962.3.sa1
Author response https://doi.org/10.7554/eLife.92962.3.sa2

## Additional files

### Supplementary files
• MDAR checklist

### Data availability
No data are newly generated or analysed during this study.

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
