## [Editor Report · eLife assessment]

This **useful** study describes the second earliest known winged ovule without a capule in the Famennian of Late Devonian. Using **solid** mathematical analysis, the authors demonstrate that three-winged seeds are more adapted to wind dispersal than one-, two- and four-winged seeds. The manuscript will help the scientific community to understand the origin and early evolutionary history of wind dispersal strategy of early land plants.

---

## [Referee Report · Reviewer #1 (Public Review)]

Summary:

Winged seeds or ovules from the Devonian are crucial to understanding the origin and early evolutionary history of wind dispersal strategy. Based on exceptionally well-preserved fossil specimens, the present manuscript documented a new fossil plant taxon (new genus and new species) from the Famennian Series of Upper Devonian in eastern China and demonstrated that three-winged seeds are more adapted to wind dispersal than one-, two- and four-winged seeds by using mathematical analysis.

Strengths:

The manuscript is well organised and well presented, with superb illustrations. The methods used in the manuscript are appropriate.

Weaknesses:

I would only like to suggest moving the "Mathematical analysis of wind dispersal of ovules with 1-4 wings" section from the supplementary information to the main text, leaving the supplementary figures as supplementary materials.

---

## [Author Response]

The following is the authors’ response to the original reviews.

The manuscript lacks the conclusion section to summarize their finding. The rebuttal is too simple to state where and in which way the authors have made their revisions. In this case, please return this revision to the authors and ask them revise their contribution carefully.

We now indicate in detail the places and the way that we make revisions. Specific revisions in sentences/words are marked with blue color in the main text where necessary. A conclusion is now provided at the end of the main text (lines 264-275). Other major revisions include:

(1) We add Fig. 5 as a new figure to reconstruct ovule structure of *Alasemenia* and to compare three- and four-winged ovules. This is followed by Fig. 6 relating to mathematical analysis.

(2) We re-organize (sequences of some) paragraphs and revise sentences in Discussion, and then divide Discussion into three parts: “Late Devonian acupulate ovules and their functions” (lines 124-150), “Late Devonian winged ovules and evolution of ovular wings” (lines 151-179), “Mathematical analysis of wind dispersal of ovules with 1-4 wings” (lines 180-262).

(3) We move “Mathematical analysis of wind dispersal of ovules with 1-4 wings” section from the supplementary information to the main text as the third part of Discussion (lines 180-262). The original paragraph headed with Mathematical analysis in Results is now modified and inserted to “Mathematical analysis of wind dispersal of ovules with 1-4 wings” section (lines 250-256). The last paragraph in the original Supplementary information is now greatly modified and presented at the end of “Mathematical analysis of wind dispersal of ovules with 1-4 wings” section (lines 256-262).

(4) With moving “Mathematical analysis of wind dispersal of ovules with 1-4 wings” section from the supplementary information to the main text, five references are accordingly added to the list (lines 278-282, 296-300, 329-330).

(5) We change the format of citing references in the main text.

We have therefore returned your manuscript to you to allow you to make the updates necessary to address the editors comments. Please ensure that you also update your preprint with the newly revised version once complete.

Many thanks for this allowance and we now make the necessary updates to address the editors’ and reviewers’ comments. At the same time, the new version is also provided as a preprint.

**Reviewer #1 (Public Review):**
Summary:Winged seeds or ovules from the Devonian are crucial to understanding the origin and early evolutionary history of wind dispersal strategy. Based on exceptionally well-preserved fossil specimens, the present manuscript documented a new fossil plant taxon (new genus and new species) from the Famennian Series of Upper Devonian in eastern China and demonstrated that three-winged seeds are more adapted to wind dispersal than one-, two- and four-winged seeds by using mathematical analysis.

Many thanks for these positive comments by the reviewer.

Strengths:The manuscript is well organised and well presented, with superb illustrations. The methods used in the manuscript are appropriate.

Many thanks for the reviewer’s positive comments.

Weaknesses:I would only like to suggest moving the "Mathematical analysis of wind dispersal of ovules with 1-4 wings" section from the supplementary information to the main text, leaving the supplementary figures as supplementary materials.

Ok, following the suggestion, we have moved this “Mathematical analysis of wind dispersal of ovules with 1-4 wings” section to the main text (lines 180-262). It now represents the third part of Discussion. The original paragraph headed with Mathematical analysis in Results is now modified and inserted to “Mathematical analysis of wind dispersal of ovules with 1-4 wings” section (lines 250-256). The last paragraph in the original Supplementary information is now greatly modified and presented at the end of “Mathematical analysis of wind dispersal of ovules with 1-4 wings” section (lines 256-262).

**Reviewer #2 (Public Review):**
Summary:This manuscript described the second earliest known winged ovule without a capule in the Famennian of Late Devonian. Using Mathematical analysis, the authors suggest that the integuments of the earliest ovules without a cupule, as in the new taxon and Guazia, evolved functions in wind dispersal.

Yes, these include our description, mathematical analysis and suggestion.

Strengths:The new ovule taxon's morphological part is convincing. It provides additional evidence for the earliest winged ovules, and the mathematical analysis helps to understand their function.

Many thanks for these positive comments of the reviewer.

Weaknesses:The discussion should be enhanced to clarify the significance of this finding. What is the new advance compared with the Guazia finding? The authors can illustrate the character transformations using a simplified cladogram. The present version of the main text looks flat.

To clarify the significance of this finding, the discussion is now enhanced in the following respects. We now re-organize the contents of Discussion and divide it into three parts. These three parts are entitled “Late Devonian acupulate ovules and their functions” (lines 124-150), “Late Devonian winged ovules and evolution of ovular wings” (lines 151-179), “Mathematical analysis of wind dispersal of ovules with 1-4 wings” (lines 180-262). The third part is transformed from the original Supplementary information.

Regarding new advance (*Alasemenia*) compared with *Guazia* and illustration of the character transformations:

(1) we now provide a new figure (Fig. 5) to reconstruct ovule of *Alasemenia* and to compare the structure of these two ovules.

(2) in the second part of Discussion, we now say “As in *Alasemenia* (Fig. 5a), the integumentary wings of acupulate ovule of *Guazia* are broad, thin and fold inwards along the abaxial side, but their numbers are four in each ovule and their free portions usually arch centripetally (Fig. 5c; Wang et al., 2022, Figure 5).”

(3) also in the second part of Discussion, we now say “Compared to *Warsteinia* with short and straight wings and *Guazia* with long but distally inwards curving wings, *Alasemenia* with longer and outwards extending wings would efficiently reduce the rate of descent and be more capably moved by wind. Furthermore, the quantitative analysis in mathematics indicates that three-winged ovules such as *Alasemenia* are more adapted to wind dispersal than four-winged ovules including *Warsteinia* and *Guazia* (see following).”

(4) in the third part of Discussion, we now say “Significantly, the maximum windward area of each wing of *Alasemenia* is greater than that of *Guazia* and *Warsteinia* with four wings. All these factors suggest that *Alasemenia* is well adapted for anemochory.”

(5) in Conclusion, we now say “Compared to Famennian four-winged ovules of *Warsteinia* and *Guazia*, *Alasemenia* with three distally outwards extending wings shows advantage in anemochory.”

**Recommendations for the authors:**

Ok, we undertake some revisions and keep some original contents.

**Reviewer #1 (Recommendations For The Authors):**
I would only like to suggest moving the "Mathematical analysis of wind dispersal of ovules with 1-4 wings" section from the supplementary information to the main text, leaving the supplementary figures as supplementary materials.

Ok, following the suggestion, we now move this “Mathematical analysis of wind dispersal of ovules with 1-4 wings” section to the main text (lines 180-262). It now represents the third part of Discussion.

**Reviewer #2 (Recommendations For The Authors):**
(1) The mathematical part as the supplement can be incorporated into the text.

Ok, following the suggestion, we now move this “Mathematical analysis of wind dispersal of ovules with 1-4 wings” section to the main text (lines 180-262). It now represents the third part of Discussion. The original paragraph headed with Mathematical analysis in Results is now modified and inserted to “Mathematical analysis of wind dispersal of ovules with 1-4 wings” section (lines 250-256). The last paragraph in the original Supplementary information is now greatly modified and presented at the end of “Mathematical analysis of wind dispersal of ovules with 1-4 wings” section (lines 256-262).

(2) The comparisons between three- or four-winged ovules are not addressed enough.

We now add Fig. 5 as a new figure. Based on this figure and revisions, the comparisons between three- and four-winged ovules now include:

a) “Their integumentary wings illustrate diversity in number (three or four per ovule), length, folding or flattening, and being straight or curving distally. As in *Alasemenia* (Fig. 5a), the integumentary wings of acupulate ovule of *Guazia* are broad, thin and fold inwards along the abaxial side, but their numbers are four in each ovule and their free portions usually arch centripetally (Fig. 5c; Wang et al., 2022, Figure 5). In contrast to *Alasemenia*, *Warsteinia* has four integumentary wings without folding and their free portions are short and straight (Rowe, 1997, TEXT-FIG. 4).” (lines 154-160).

b) “Furthermore, the quantitative analysis in mathematics indicates that three-winged ovules such as *Alasemenia* are more adapted to wind dispersal than four-winged ovules including *Warsteinia* and *Guazia* (see following).” (lines 166-168).

c) “The relative wind dispersal efficiency of three-winged seeds is obviously better than that of single- and two- winged seeds, and is close to that of four-winged seeds (Fig. 6). In addition, three-winged seeds have the most stable area of windward, which also ensures the motion stability in wind dispersal. Significantly, the maximum windward area of each wing of *Alasemenia* is greater than that of *Guazia* and *Warsteinia* with four wings.” (lines 256-261).

d) “Compared to Famennian four-winged ovules of *Warsteinia* and *Guazia*, *Alasemenia* with three distally outwards extending wings shows advantage in anemochory.” (lines 272-274).

(3) The significance of this finding should be well summarized with solid evidence.

It has been summarized in Abstract (lines 19-28) and is now further summarized especially in the newly provided Conclusion (lines 264-275).